# Brain Drain out of the Blue: Pollution-Induced Migration in Vietnam

**DOI:** 10.3390/ijerph19063645

**Published:** 2022-03-18

**Authors:** Quy Van Khuc, Minh-Hoang Nguyen, Tam-Tri Le, Truc-Le Nguyen, Thuy Nguyen, Hoang Khac Lich, Quan-Hoang Vuong

**Affiliations:** 1Center for Economic Development Studies and Faculty of Political Economy, VNU University of Economics and Business, Vietnam National University, Hanoi 100000, Vietnam; qvkhuc@vnu.edu.vn (Q.V.K.); trucle@vnu.edu.vn (T.-L.N.); 2Centre for Interdisciplinary Social Research, Phenikaa University, Yen Nghia Ward, Ha Dong District, Hanoi 100803, Vietnam; tri.letam@phenikaa-uni.edu.vn; 3Vietkaplab, Hanoi 100000, Vietnam; diepnn1519@gmail.com; 4Personnel Department, University of Economics and Business, Vietnam National University, Hanoi 100000, Vietnam; lichhk@vnu.edu.vn

**Keywords:** air pollution, brain drain, emigration, Bayesian mindsponge framework (BMF)

## Abstract

Air pollution is a major problem that severely affects the health of inhabitants in developing countries’ urban areas. To deal with the problem, they may consider migration to another place as an option, which can result in the loss of skillful and talented workforces. This situation is called the brain drain phenomenon. The current study employed the Bayesian mindsponge framework (BMF) on the responses of 475 urban inhabitants in Hanoi, Vietnam—one of the most polluted capital cities in the world—to examine the risk of losing talented workforces due to air pollution. Our results show that people with higher educational levels are more likely to have intentions to migrate both domestically and internationally due to air pollution. Regarding the domestic migration intention, younger people and males have a higher probability of migrating than their counterparts. Age and gender also moderate the association between educational level and international migration intention, but their reliability needs further justification. Based on these findings, we suggest that environmental stressors caused by air pollution can influence citizen displacement intention on a large scale through the personal psychological mechanism of cost-benefit judgment. Due to the risk of air pollution on human resources, building an eco-surplus culture is crucial for enhancing environmental and socio-economic resilience.

## 1. Introduction

In the age of industrialization and a fierce race for material wealth, air pollution has emerged as a worldwide concern directly impacting human beings’ future survival. Air pollution brings along many issues related to the environment and human health. According to the World Health Organization (WHO), places that fail to meet WHO’s air quality standards accommodate approximately 91% of the world population. Air pollution is a root cause of about seven million deaths annually [1]. Without considerable endeavors to tightly control pollution effects, estimated figures for fatalities caused by ambient air pollution may range from 6,000,000 to 9,000,000 by 2060 [2]. It is reported that regardless of similar effects of ambient air pollution on both first-world and third-world countries, middle- and low-income countries have been subject to a higher severity level of air pollution [1]. Developing countries are reported to have fewer provisions of preventive health services [3], higher children exposure to air pollution [4,5], and higher maternal exposure during pregnancy [6] than their developed counterparts.

Air pollution has a positive relationship with cardiovascular diseases [7] and chronic respiratory diseases [8,9], such as asthma, pneumonia [10], and lung cancer [11], and decreased life expectancy [12,13]. The elderly, children, and patients with pre-existing conditions related to respiratory and cardiovascular diseases face great danger from just a small number of air toxicants [14]. Overexposure to air pollutants causes molecular and cell toxicity, thereby facing an increased risk of cancer development in the long run [15]. It is important to note that air pollution also poses constant threats to human mental health. PM levels are linked with deteriorated mental health, and those exposed to PM2.5 for a long time are prone to depressive symptoms [16]. Moreover, research suggests that an air-polluted environment was also positively associated with declining cognitive ability [17,18], psychological disorders [19], distress [20], and even suicide [21,22]. According to Mohai et al. [23], a polluted learning environment was likely to negatively impact children’s health and jeopardize their academic performance, evidenced by a low attendance rate and a low success rate in the state’s standard tests. Notably, air pollution impairs cognitive ability and increases the risks of dementia among adults [24] and the elderly [25,26].

Far-fetching ramifications for polluted air calls for government intervention to guarantee community well-being. China [27], European countries [28,29], and the U.S. [30] have enforced regulations on air pollution and mitigation policies. Although there are a variety of regulations in place, previous research demonstrates mixed results of the limited effectiveness of environmental regulations and unachieved desired goals of air pollution control [31,32]. Although there are a variety of regulations in place, a race for economic growth has exacerbated an already bad air pollution situation. Therefore, apart from governmental endeavors, individuals actively seek several ways to keep pollution effects to a minimum. Residents may buy particulate-filtering face masks [33], favor short-term avoidance travels [34] and sedentary activities [35], and reduce their time on outdoor activities [36]. Another potential measure to counter negative effects is cross-national movements, which means that individuals tend to move from heavily polluted countries to better ones. The brain drain effect occurs when high-quality workforces leave their current areas leading to disruption in the distribution of intellectual capital. The phenomenon of brain drain has been the target of scientific research for over 50 years; this research branch is growing recently, and the phenomenon is studied in many different disciplines [37]. According to the Fragile States Index [38], from 2006 to 2021, the situation of brain drain in Vietnam has improved, with a declined Human Flight and Brain Drain Index and an overall fragility rank of 114th. However, the index is still worrying compared to other countries in the region (5.3 compared to 4.5 and 1.9 of Malaysia and Singapore, respectively).

Regarding emigration behavior, potential determinants lie with economic growth in the destination countries, namely income level and employability; thus, college-educated individuals are more inclined to emigrate in search of economic opportunities [39]. Many empirical studies [40,41,42,43] dispel the misconception of a linear relationship between economic development and emigration rates and confirm an inverted U-shaped relationship between these two factors. According to a 2017 estimation, the proportion of total international migrants was 3.5% of the total world population in 2050 [44]. Still, in 2019, the number of international migrants increased by 3.5%, with a total of 272 million people, most of whom were 20–64-year-olds (roughly 74%) [45].

The brain drain effect in our investigation seems to be “out of the blue” on two different aspects: it is induced by the desire to escape one’s polluted living place, and the mechanism behind this ideation appears unclear. Previous research found that the brain drain hypothesis was not validated in terms of domestic migration intention among urban inhabitants in Vietnam [46]. However, from the mindsponge information-processing perspective, we anticipate that the brain drain effect may be more complex than a linear relationship. The cost-benefit judgments of an individual about migration incorporate not only educational level but also other factors associated with their characteristics, such as age and gender. Thus, our research investigates whether the associations between educational level and domestic and international migrations are moderated by age and gender.

## 2. Materials and Methods

### 2.1. Model Construction

In a former study using the same dataset on the psychological process of domestic migration intention due to air pollution [46], the brain drain effect was not found. Still, gender and age factors are shown to have a clear influence. The fact that the brain drain effect was not found in the mentioned study does not seem to be consistent with other studies in other countries that confirm the effect’s existence. Thus, we postulate that the direct association between educational level and migration intention appears unclear in that study because it was analyzed as a linear relationship instead of a non-linear one.

To further examine the above relationship and shed light on the presented inconsistency, we employed the Bayesian mindsponge framework (BMF) [47,48]. BMF utilizes the mindsponge mechanism of information processing as the theoretical foundation and employs Bayesian inference to construct statistical analysis. Since its original conception [49,50], the mindsponge mechanism has been expanded and proven to be effective in explaining how *ideation* occurs in the human mind based on subjective cost-benefit evaluation [48,51]. It is also essential when dealing with complex information processes [52]. 

Migration is an important decision in a person’s life. Thus, the information process leading to such an impactful decision requires multifaceted evaluation. Apart from educational level, other socio-economic factors, such as the age and gender of inhabitants, might simultaneously influence migration intention. It is necessary to view the association between educational attainment and migration as a non-linear relationship with interactions among many factors.

Based on this reasoning, in the present study, we examine how educational level and its interactions with gender and age influence the intention to migrate domestically due to air pollution, as shown in Model 1.
(1)MoveCity ~ α+Education+Education×AgeGroup+Education×Gender

Additionally, we want to examine how the same set of factors influences international migration intention. The differences in the corresponding subjective cost-benefit judgments between domestic and international migration can provide more insights into the rationale behind these target-specific decisions. It is presented in Model 2, where interaction terms include *Education*, rendering the specification non-linear by nature.
(2)MoveCountry ~ α+Education+Education×AgeGroup+Education×Gender

### 2.2. Materials

The current study employed the samples retrieved from two random-sampling datasets that were generated by Khuc et al. [53] and Vuong et al. [54]. The data collections ranged from November to December of 2019. At the time of collection, Hanoi—the second populous city in Vietnam—was heavily affected by air pollution. Among the most polluted capital cities worldwide, its pollution level was ranked 7th.

To ensure the data collection quality, Khuc, Phu, and Luu [53] recruited and trained collectors to ensure sufficient knowledge and skills for the task. Two pilot surveys were also conducted for quality checks.

We employed five variables for constructing two models in the current study: three predictor variables and two outcome variables. The description of each variable is presented in Table 1.

### 2.3. Methods and Validation

Following the BMF [47], the Bayesian method was employed in this study because its properties fit the complexity of models constructed using the mindsponge mechanism. Since the human mind’s psychological processes are multiplex, constructing parsimonious models can help improve precision and predictability. The most prominent feature that makes the Bayesian inference approach suitable for parsimonious models is that it treats all properties probabilistically, including unknown parameters. Furthermore, recent Bayesian analysis is aided by the Markov chain Monte Carlo (MCMC), which enables the estimation of models with high complexity, such as the non-linear relationships examined in this study. The feature of stochastic processes of Markov chains helps generate a large number of iterative samples and data points, which fits complex models effectively. The technique, therefore, helps meet the large-sample-size requirement for sound estimation of non-linear relationships [55].

Science faces a reproducibility crisis that hinders the reproduction of studies, including psychology [56] and social sciences [57]. The main reason for the crisis is pointed out to be the wide sample-to-sample variability in the *p*-value [58]. Hence, avoiding the replication crisis by limiting the over-reliance on the *p*-value is another advantage that Bayesian probability was employed in the present study besides its high compatibility with the mindsponge framework. Moreover, due to the MCMC technique’s integration and symmetry assumption independence, Bayesian analysis can precisely estimate the small samples at hand and asymmetric distributions [59,60].

After the simulated results were obtained, three validation steps were conducted. First, the Pareto smoothed importance-sampling leave-one-out cross-validation (PSIS-LOO) diagnostic was used to check the models’ goodness-of-fit with the current samples [61]. Specifically, if the *k* values shown on the plot are below the threshold value of 0.5, the model can be deemed a good fit with the data. Next, diagnostic statistics and plots were employed to validate whether the stochastic simulation process has the Markovian property—in other words, the simulated samples are not autocorrelated. The effective sample size (n_eff) and the Gelman shrink factor (Rhat) are parts of the diagnostic statistics, while the trace, Gelman, and autocorrelation plots are the diagnostic plots. Details of the diagnostic statistics and plots are presented and explained in the Results section. Finally, “prior-tweaking” was performed to check the sensitivity of the brain drain effect (the association between educational level and migration intention) toward prior modification. To elaborate, besides estimating the posteriors using uninformative priors, we also estimated the posteriors using informative priors representing our belief and disbelief on the brain drain effect. To incorporate our belief into the model, we set the prior *Education* parameter as a normal distribution, with mean at 1 and standard deviation at 0.5: norm (1, 0.5). In contrast, the prior with normal distribution with mean at 0 and standard deviation at 0.5 represents our disbelief: norm (0, 0.5).

In the current study, we use the bayesvl R package to conduct Bayesian analysis due to its three prominent features: (1) easy-to-use operation; (2) clear visualization; (3) good cost-effectiveness [62]. For transparency and later replication or cross-checking purposes [63,64], the dataset, data description, and code snippets of the current study’s analysis were all deposited on The Open Science Framework (see data availability statement).

## 3. Results

We employed Bayesian analysis with 5000-iteration MCMC simulation (2000 warm-up iterations and four Markov chains) on two models in the current study. The simulated results and technical validity of the models are presented accordingly. In the dataset, the male and female percentage of respondents is 54.53% and 45.26%, respectively. The proportion of the 10–30 age group is 61.47%—the highest proportion in the sample. About 5% of respondents reported having intentions of migrating to another city, and approximately 7.6% of respondents had intentions of migrating to a foreign country due to air pollution.

### 3.1. Model 1: Domestic Migration Intention

Model 1 investigates the predictions of respondents’ education level and its interactions with their age group and their gender towards migration intention to a less polluted province. For clarity, we show Model 1′s logical network in Figure 1.

Figure 2 indicates that all *k* values are below the threshold of 0.5, so Model 1 has a high goodness-of-fit with the data.

The diagnostic statistics portray a good convergence of the model’s Markov chains because the effective sample sizes (n_eff) are larger than 1000, and Gelman shrink factor (Rhat) statistics are equal to 1 (see Table 2). The trace plots, autocorrelation plots, and Gelman plots also confirmed the good convergence of the model.

Figure 3 demonstrates all posterior parameters’ trace plots. The posterior value of each parameter is represented on the *y*-axis, while the *x*-axis indicates the iteration order. Four colored lines in the middle are the Markov chains. If the Markov chains fluctuate around a central equilibrium, they can be deemed good-mixing and stationary. These two characteristics are signals that the Markovian property is held.

The Gelman plots of Model 1′s parameters are presented in Figure 4. The shrink factor (or Gelman factor), used to assess the relativeness of the variance between Markov chains and the variance within chains, is presented on the *y*-axis. There is no divergence among Markov chains, as the shrink factors of all parameters drop quickly to 1 during the warm-up iterations (before the 2000th iteration). Therefore, the Markov chains obtain good convergence.

In the next step, we use the Markov chains’ autocorrelation levels to validate Model 1′s convergence (see Figure 5). The *x*-axes of the plots present the Markov chains’ lag, while the *y*-axes illustrate the average level of autocorrelation of each chain. It can be seen that the average autocorrelation level declines quickly before the fifth lag, indicating that all parameters acquire a healthy number of effective samples (see Figure 5). The autocorrelation plots also confirm the good convergence of Model 1′s Markov chains.

From the simulated posterior results using uninformative priors of Model 1, we found that the education level of respondents was positively associated with the intention to migrate to another city ($\mu_{Education}=0.35$ and $\sigma_{Education}=0.25$). This result confirms our assumption that the higher education level an individual has, the more likely it is that they will intend to migrate to another city. Regarding the interaction with gender, males are found to be positively associated with a higher level of domestic migration ($\mu_{Gender\_Education}=0.14$ and $\sigma_{Gender\_Education}=0.12$). However, the interaction with age groups produces a negative value, suggesting old age lessens the effect of education level on domestic migration intention ($\mu_{AgeGroup\_Education}=-0.11$ and $\sigma_{AgeGroup\_Education}=0.05$). The estimated posteriors using the priors demonstrating our belief and disbelief on the brain-brain effect only slightly differ from the estimated result using uninformative prior. Moreover, the tendency of the *Education* parameter also does not change. Based on these points, we can say that the posterior of *Education* remains insensitive even when we have prior belief or disbelief toward the brain drain effect.

Figure 6 shows the interval plot of the posterior distributions of Model 1′s parameters. The *x*-axis of the plot presents the probability distribution of parameters. Most of the distribution of *Education* lies on the positive side of the axis, indicating a highly reliable positive association between *Education* and *MoveCity*. The distribution of *Gender* × *Education* is mostly located on the positive sides, implying that the male gender had a positive influence (moderate confidence) toward the effect of education level on domestic migration intention. In terms of age, most of the distribution of *AgeGroup* × *Education* lies on the negative side, implying old age was negatively associated with the intention to migrate to a less polluted province.

The logit model below calculates the probability of domestic migration intention among urban inhabitants. To calculate the probability, we selected the parameters’ mean values estimated using uninformative priors because they have the highest probability of happening.
(3)$$\begin{array}{c} ln\frac{\pi_{domestic\;migration}}{\pi_{no\;domestic\;migration}}=-2.56+0.35\times Education-0.11\times AgeGroup\times Education\\ +0.14\times Gender\times Education \end{array}$$

Based on this model, the probability of domestic migration intention of male urban inhabitants who obtained Bachelor’s degrees and are around 31–40 years old can be calculated as follows:(4)$$\pi_{domestic\;migration}=\frac{e^{-2.56+0.35\times 3-0.11\times 3\times 3+0.14\times 1\times 3}}{1+e^{-2.56+0.35\times 3-0.11\times 3\times 3+0.14\times 1\times 3}}=0.1111=11.11\%$$

The calculation was applied similarly to other scenarios. The probabilities of male and female urban inhabitants are shown in Figure 7a,b, respectively. It is readily seen from Figure 7a,b that urban people with higher educational levels generally have higher probabilities of migrating domestically due to air pollution. However, male inhabitants older than 50 are less likely to have migration intentions. In the case of female inhabitants, people older than 30 years old do not have an increased likelihood of domestic migration if they obtain a higher educational level. Female inhabitants older than 40 have even less probability of domestic migration if they have higher educational levels.

### 3.2. Model 2: International Migration Intention

The second model examines the effects of respondents’ education level and its interaction with their age group and gender on international migration intentions due to air pollution concerns. The logical network of Model 2 can be illustrated in Figure 8.

The *k* values on the PSIS diagnostic plot are lower than 0.5, showing a high goodness-of-fit of the model (see Figure 9).

Model 2′s statistical results are well validated. All n_eff values are greater than 1000, and Rhat values are equal to 1. The convergence of Model 2′s Markov chains is again confirmed through other visual diagnostic methods, such as the trace plots (see Figure A1), the Gelman plots (see Figure A2), and the autocorrelation plots (see Figure A3).

From the simulated posterior results using uninformative priors of Model 2 in Table 3, we found that the education level of respondents was positively associated with the intention to migrate to a less polluted country ($\mu_{Education}=1.10$ and $\sigma_{Education}=0.33$). This result is consistent with Model 1′s results. After applying the “prior-tweaking” technique, it can also be seen that the association between educational level and international migration intention only changes slightly, validating the robustness of the brain drain phenomenon in international migration.

However, the moderation effect of gender is different from Model 1′s result. Males were found to have a lower probability of international migration intention, but this result has a high degree of deviation ($\mu_{Gender\_Education}=-0.11$ and $\sigma_{Gender\_Education}=0.13$). The moderation effect of *AgeGroup* on the relationship between *AgeGroup* and *MoveCountry* also has a high degree of deviation ($\mu_{AgeGroup\_Education}=-0.03$ and $\sigma_{AgeGroup\_Education}=0.05$). All coefficients’ probability distributions are presented in Figure 10. 

Using the similar logit model above, we could also calculate the probability of international migration intention among urban inhabitants. The model is shown below:(5)$$\begin{array}{c} ln\frac{\pi_{international\;migration}}{\pi_{no\;international\;migration}}=-5.29+1.10\times Education-0.03\times AgeGroup\times Education\\ -0.11\times Gender\times Education \end{array}$$

Based on this model, the probability of international migration intention of female urban inhabitants who earned the Master’s degrees and are around 31–40 years old can be calculated as follows:(6)$$\pi_{international\;migration}=\frac{e^{-5.29+1.10\times 4-0.03\times 3\times 4-0.11\times 0\times 4}}{1+e^{-5.29+1.10\times 4-0.03\times 3\times 4-0.11\times 0\times 4}}=0.2227=22.27\%$$

The calculation was applied similarly to other scenarios. The probabilities of male and female urban inhabitants are shown in Figure 11a,b, respectively. As can be seen from the graphs, the probability of international migration intention increases as the educational level increases. Although younger people have a higher probability of migrating internationally, the change is not minimal. Meanwhile, gender has a greater effect on migration intention probability when the educational level is higher, especially after getting a Master’s or Doctoral degree.

## 4. Discussion

The current study was conducted using the Bayesian mindsponge framework (BMF) to examine the possibility of brain drain effect due to air pollution and the moderation effects of age and gender among 475 Vietnamese urban inhabitants. The models were constructed based on the mindsponge information processing mechanism and estimated using Bayesian inference. We found that the brain drain effect may happen domestically and internationally, as the respondents’ educational level is positively associated with the domestic and international migration intentions. The age of the respondents negatively moderates the associations, and the moderation effect on the relationship between education and international migration is situation specific.

In the current study, the domestic and international brain drain phenomena were found to happen depending on the inhabitants’ age. There are several explanations behind this result. Given that migration, regardless of domestic or international migration, is an important decision that results in long-term consequences, during the cost-benefit evaluation process, an individual needs to consider the possibilities of actualizing the migration intention and sustaining a living in the destination. Thus, having the ability to find a job in the targeted location may drive the moving decision. On the one hand, the highest attained educational level, to an extent, can represent the respondents’ working ability and improve their confidence in job finding. On the other hand, acquiring a higher educational level provides the respondent with a wider range of work choices in the new place, reducing the perceived risk of unemployment.

Nevertheless, educational level is only one of many factors that must be considered when pondering upon a lifetime decision, like migration. Vuong, Le, Nguyen, and Nguyen [51] suggested that some traits of *Homo oeconomicus*, who always acts to maximize the economic benefits, might also affect the inhabitants’ migration consideration due to air pollution. Some distance costs (e.g., finance, culture shock) exist when migrating to a new place [65,66], so younger, highly educated people would have more time to regain such loss. These people still have an expected long future ahead, and the economic opportunities in the new living environment are perceived to outweigh short-term displacement costs. On the contrary, older, highly educated people have limited time and are likely more prone to acculturative stress. This group does not perceive the opportunities in a new place as attractive as the young group does due to their age, and thus, they have weaker economic incentives for migration.

Moreover, older, highly educated people are usually those that have already accumulated a certain amount of fortune in their current city. In comparison, although they are better educated, younger people still need time to accumulate wealth. As a result, older, highly educated people’s perceived moving costs can be greater than those of younger, highly educated ones.

Besides age, gender is also an element that moderates the relationship between the respondent’s educational level and migration intention. Highly educated male respondents are more likely to have domestic migration ideation than female counterparts. Moreover, as shown in Figure 7B, female urban people older than 30 years old do not have domestic migration intention regardless of educational levels. Those older than 40 even witness a decline in domestic migration probability if they have higher educational levels. The moderation effect of gender on the association between educational attainment and domestic migration intention might be explained by the influence of Confucianism ideology in Vietnamese households [67,68]. Vietnamese women over the age of 30 are likely to get married, have a family with children, and tend to be under higher family-related pressures due to cultural expectations about the female responsibility of taking care of young children. Women with higher educational levels also have a higher chance of acquiring a better job, so they might perceive the domestic migration intention to be more costly.

However, the moderation effect of gender on the association between educational level and migration intention is quite different in the case of international migration intention. Specifically, female inhabitants are more likely to have an international migration intention than male counterparts at the same educational level. The difference is even clearer at the Master and Doctoral levels (see Figure 11a,b).

The conflicting effects of gender on the associations between educational levels and domestic and international migration intentions require further studies from various angles, including socio-cultural and economic aspects. They also suggest that the effect of gender should not be overgeneralized but should be further examined using other methodologies.

It is conclusive that educational attainment is a crucial factor in the cost-benefit judgments of urban inhabitants about domestic and international migration. Still, it is not the only important element. The individual’s evaluation of migration ideation due to air pollution is a multiplex process comprising multiple socio-cultural and economic factors. Because of this multiplexity, earlier research could not find the linear association between the educational level and domestic migration intention among urban residents in Vietnam [46].

Still, the risk is clear. Environmental degradation can lead to a potential loss of valuable human resources, such as skilled workers and college graduates, in the source region or country. Our findings show a brain drain effect (people with higher educational levels are more likely to migrate) due to air pollution perception among Hanoi residents. The effect was found concerning domestic as well as international migration intentions. This result is consistent with former studies providing evidence on the brain drain effect for both domestic and international migration [39,69,70,71,72,73].

We recognize that the brain drain effect is even more serious in the case of Vietnam, and Hanoi in particular, because those being more likely to migrate are not just talented but are also young and still at their prime regarding working capacity. Such realization again emphasizes the importance of long-term consideration in environmental policies because pollution can negatively impact the socio-economic state of a region or country, not only now but also far into the future. Therefore, building an “eco-surplus culture”, especially among private sectors, is not only for environmental sustainability but also for socio-economic resilience [74].

## 5. Conclusions

Due to concern about air pollution, urban residents may try to migrate. The brain drain phenomenon occurs when highly educated people leave their living areas. Our study provided evidence that this happens on both national and international scales to the population of Hanoi, Vietnam. The brain drain effect can be considered “out of the blue” due to two different reasons from two perspectives. Regarding the residents’ perspectives, they want to relocate out of the undesirable polluted zone. As for researchers’ perspectives, the effect is found to be non-linear and influenced by many economic socio-cultural factors. The mechanism of migration ideation is multiplex and requires careful consideration of socio-economic aspects as well as local cultural values. None of these can be under the control of any resident, and the level of pollution that leads to a migration intent could hardly be known.

The present study presents an effective approach for exploring the psychological processes behind pollution-induced migration intention based on the mindsponge information processing framework, which can be highly applicable in further studies in other developing countries. However, it also has certain limitations: the sample only includes residents of Hanoi city, and the number of residents holding Master’s and Doctoral degrees is quite small (regarding this aspect, the use of the MCMC technique in analysis helps increase statistical accuracy). Furthermore, our study could provide insights into the underlying mechanism behind the brain drain phenomenon, but it could not indicate where the migration flows will go due to data limitations. Therefore, additional studies regarding the migration flow due to the brain drain effect should be conducted using a more detailed dataset and spatial econometrics.

## Figures and Tables

**Figure 1 ijerph-19-03645-f001:**
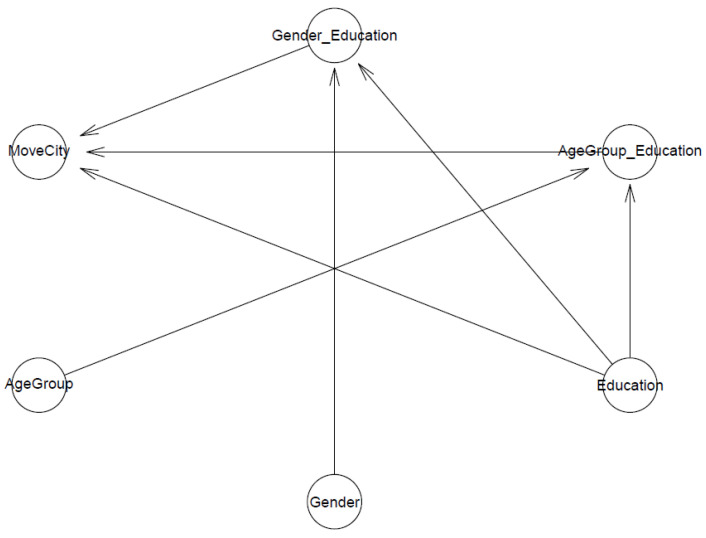
Model 1′s logical network.

**Figure 2 ijerph-19-03645-f002:**
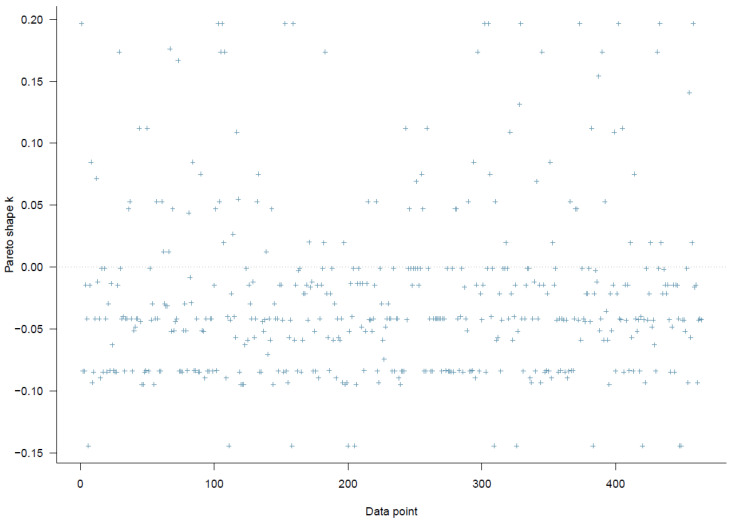
Model 1′s PSIS diagnostic plot.

**Figure 3 ijerph-19-03645-f003:**
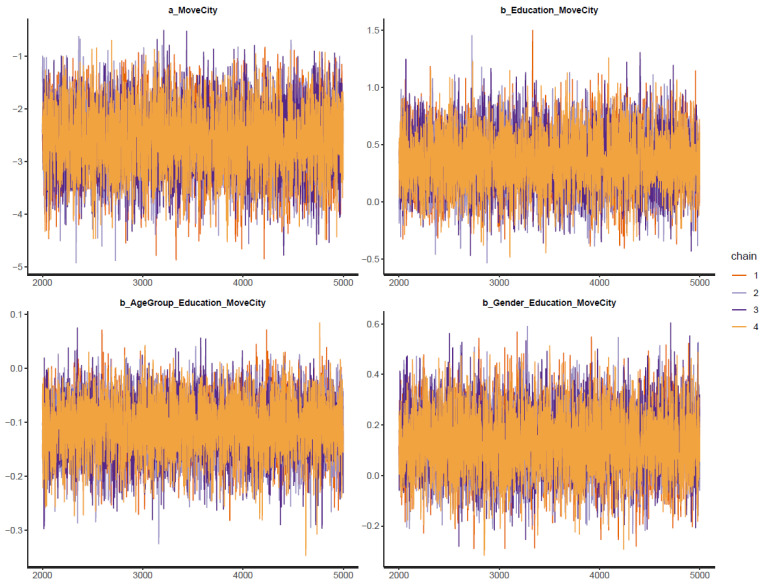
Trace plots for Model 1′s posterior parameters.

**Figure 4 ijerph-19-03645-f004:**
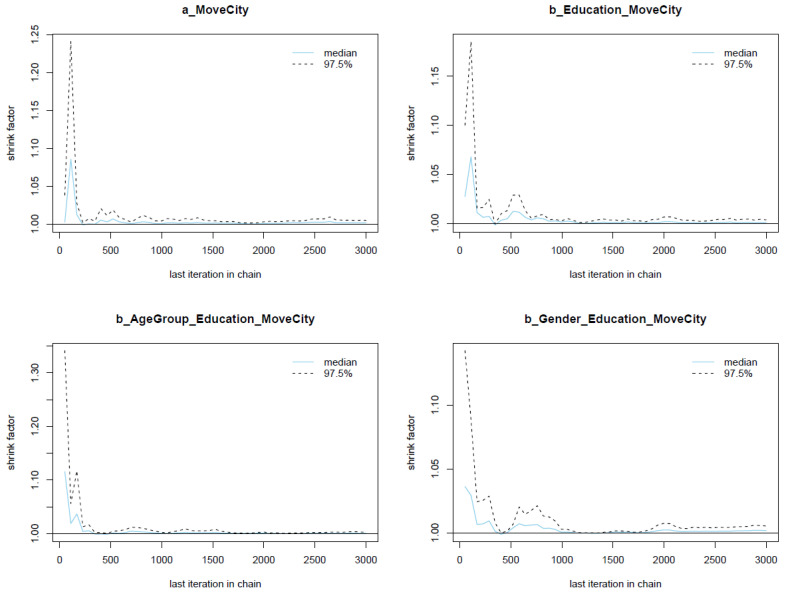
Gelman plots for Model 1′s posterior parameters.

**Figure 5 ijerph-19-03645-f005:**
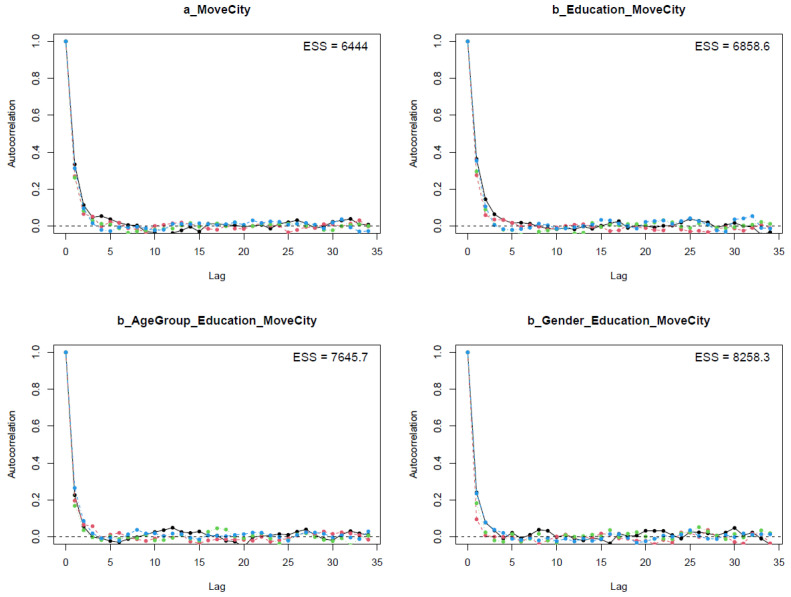
Autocorrelation plots for Model 1′s posterior parameters.

**Figure 6 ijerph-19-03645-f006:**
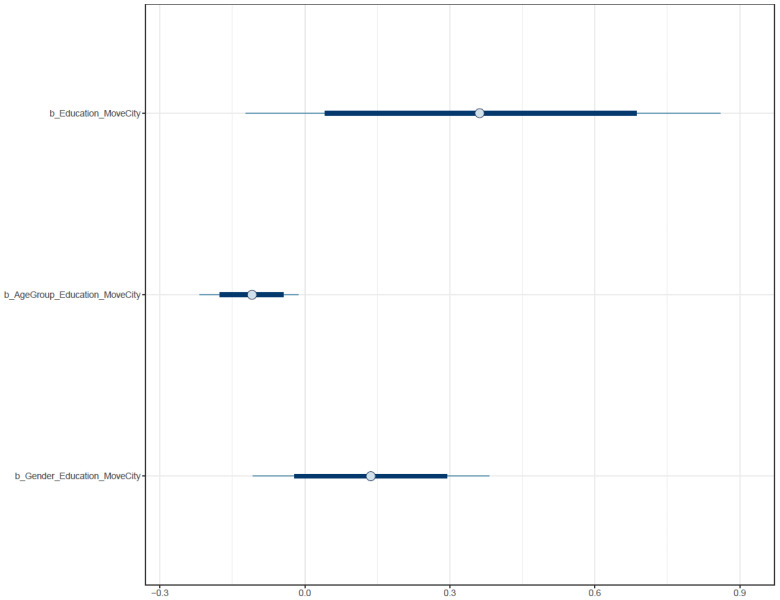
Distribution of Model 1′s posterior coefficients estimated with uninformative priors.

**Figure 7 ijerph-19-03645-f007:**
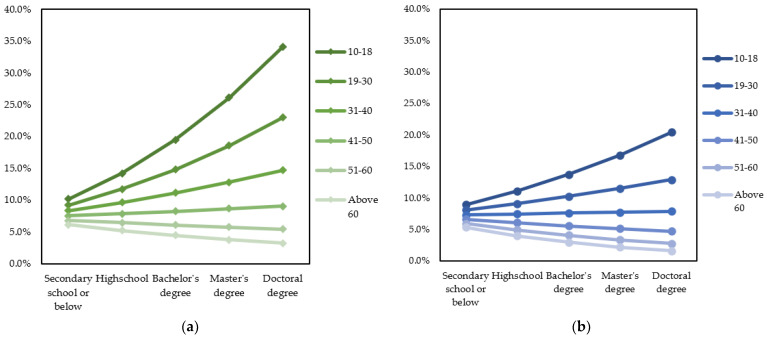
Probabilities to have domestic migration intention based on educational level and age group. (**a**) Male urban inhabitants; (**b**) female urban inhabitants.

**Figure 8 ijerph-19-03645-f008:**
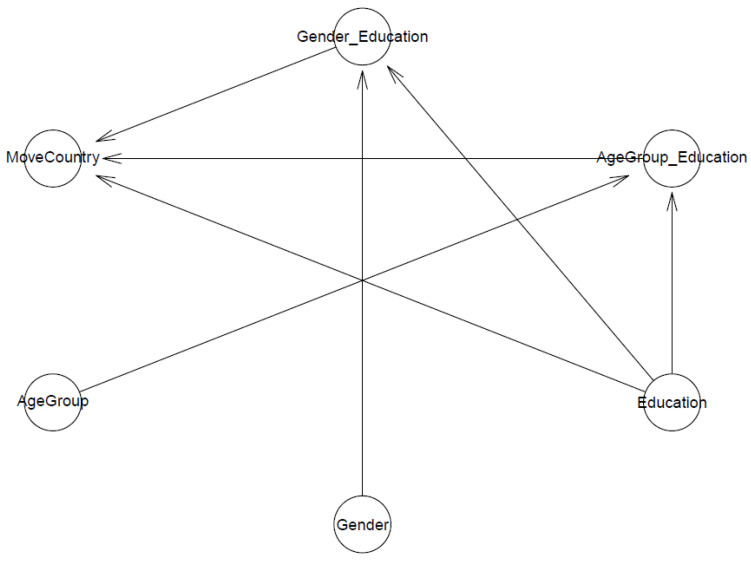
Model 2′s logical network.

**Figure 9 ijerph-19-03645-f009:**
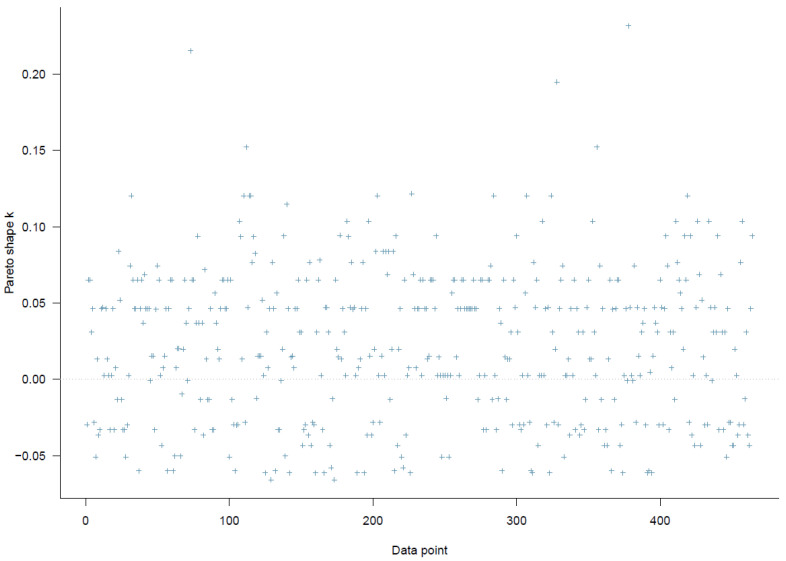
Model 2′s PSIS diagnostic plot.

**Figure 10 ijerph-19-03645-f010:**
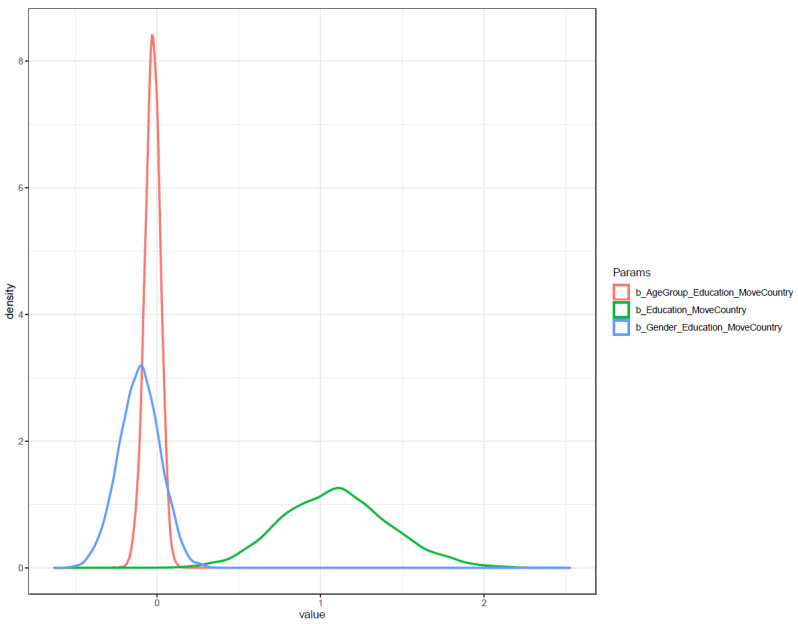
Distributions of Model 2′s posterior coefficients with uninformative priors on a density plot.

**Figure 11 ijerph-19-03645-f011:**
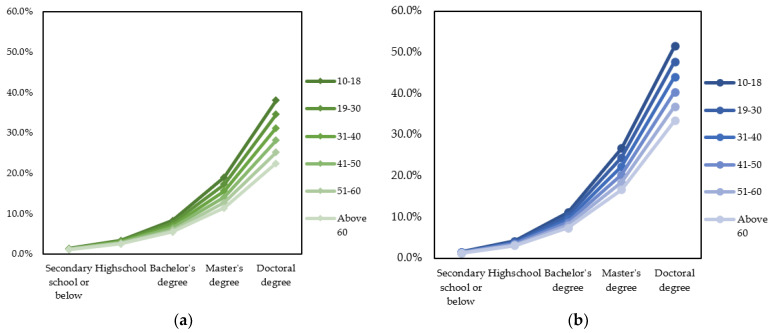
Probabilities to have international migration intention based on educational level and age group. (**a**) Male urban inhabitants; (**b**) female urban inhabitants.

**Table 1 ijerph-19-03645-t001:** Variable description.

Variable	Meaning	Type of Variable	Value
*MoveCity*	The intention of moving their family and work to a less polluted province due to the consideration of air pollution	Binary	Yes = 1 No = 0
*MoveCountry*	The intention to move their family and work to a less polluted foreign country due to the consideration of air pollution	Binary	Yes = 1 No = 0
*Gender*	Gender of the respondent	Binary	Male = 1 Female = 0
*Education*	The highest achieved level of education of the respondent	Continuous	Secondary school or below = 1 Highschool = 2 Technical school/College degree and Bachelor’s degree = 3 Master’s degree = 4 Doctoral degree = 5
*AgeGroup*	The age group that the respondent belongs to	Continuous	From 10 to 18 = 1 From 19 to 30 = 2 From 31 to 40 = 3 From 41 to 50 = 4 From 51 to 60 = 5 Above 60 = 6

**Table 2 ijerph-19-03645-t002:** Model 1′s simulated posterior coefficients.

Parameters	Uninformative Priors				Belief in the Brain Drain *N* (1, 0.5)				Disbelief in the Brain Drain *N* (0, 0.5)			
	Mean	Standard Deviation	n_eff	Rhat	Mean	Standard Deviation	n_eff	Rhat	Mean	Standard Deviation	n_eff	Rhat
*Constant*	−2.56	0.64	5921	1	−2.84	0.61	6992	1	−2.43	0.59	6312	1
*Education*	0.35	0.25	5188	1	0.50	0.22	7131	1	0.28	0.23	6314	1
*AgeGroup_Education*	−0.11	0.05	5837	1	−0.12	0.05	8089	1	−0.10	0.05	7235	1
*Gender_Education*	0.14	0.12	7214	1	0.11	0.12	8150	1	0.15	0.12	7512	1

**Table 3 ijerph-19-03645-t003:** Model 2′s simulated posterior coefficients.

Parameters	Uninformative Priors				Belief in the Brain Drain *N* (1, 0.5)				Disbelief in the Brain Drain *N* (0, 0.5)			
	Mean	Standard Deviation	n_eff	Rhat	Mean	Standard Deviation	n_eff	Rhat	Mean	Standard Deviation	n_eff	Rhat
*Constant*	−5.29	0.91	5312	1	−5.20	0.81	6027	1	−4.51	0.75	6060	1
*Education*	1.10	0.33	5188	1	1.07	0.28	6051	1	0.77	0.26	5942	1
*AgeGroup_Education*	−0.03	0.05	7837	1	−0.03	0.05	7652	1	−0.01	0.05	7652	1
*Gender_Education*	−0.11	0.13	8214	1	−0.10	0.13	8746	1	−0.07	0.13	8154	1

## Data Availability

The data and code that support the findings of this study are available on The Open Science Framework for later replications (https://osf.io/us5tr/; accessed on 14 February 2022).

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
