# Peer review of "Brain Drain out of the Blue: Pollution-Induced Migration in Vietnam"

_ijerph, 2022, doi:10.3390/ijerph19063645_

Round 1

Reviewer 1 Report

It is a good paper, addressing the issue of Brain drain in Vietnam due to air pollution. While the quantitative analysis is compelling and the findings are interesting, the authors have to convince the potential readers that their empirical results, drawn from a country- specific environment, can be relevant for other countries as well.

Also, the paper needs a proper Conclusion section (currently there is none), highlighting the main findings and their policy implications, authors’ original contribution to knowledge, the research limits and potential future research directions.

On a minor note, I recommend additional proofreading of the paper because some paragraphs lack clarity.

Author Response

Thank you for your comments! Kindly see the attached file for our responses.

Reviewer 2 Report

General remarks

The manuscript aims to determine the influence of air pollution on (internal and/or external) migration intentions. For this, the particular case of respondents from the city of Hanoi, Vietnam, to a survey, whose responses were the target of a Bayesian Mindsponge Framework, is considered. The authors state that it is possible to conclude that individuals with higher education are those who are more likely to migrate, either internally or externally, towards less polluted locations.

Specific remarks

Overall, I enjoyed reading the manuscript, which addresses a topic relevant and appropriate for the Journal. Even so, there are specific aspects that I strongly recommend that they be reviewed. They are as follows:

  • First of all, I think that the title of the manuscript should be revised, so that the reader immediately knows which developing country is being considered;
  • Secondly, I believe that the manuscript should undergo a second reading, in order to make it clearer and with fewer typos. For example, on page 3, 3rd paragraph, the word "intentions" appears by itself. The final sentence in this paragraph also seems grammatically incorrect. For another example, in the caption of table 2 (page 6) the word "posterio" appears. Still for another example, the title of the reference Nguyen, M. et al. (2021), in the list of references, is incomplete. Still for some possible flaws, can the authors explain why the percentage of male and female respondents does not add up to 100%? See “In the dataset, the male and female percentage of respondents is 54.53% and 45.21%, respectively.” (page 5);
  • Regarding the methodology, I recommend that the authors show that it would not change if the education and age group variables were considered discrete variables, as seems to be the case, and not continuous variables (see table 1 on page 4 );
  • As it seems to me to be desirable, I also recommend that the concluding section contains the limitations of the study, eventually to be considered in future works.

Author Response

(The authors gave the same response as above.)

Reviewer 3 Report

Brain drain is an interesting topic, which needs to be renewed with other views, therefore the theme of the article is pertinent. However, some improvements are needed:
1) It is necessary to detail in the introduction and discussion, the meaning of using the expression "out of the blue".
2) It is necessary to refer in the introduction and at least consider brain circulation as a line of future research, since it is a more contemporary behavior (trend within brain drain studies), than the brain drain without return. Chequear: https://doi.org/10.3390/su13063195
3) It is necessary to discuss the results against the (essentially international) brain drain phenomenon in the context of Vietnam. For this it is necessary to observe the evolution of the brain drain phenomenon in that country in the Fragile States Index (Component E3: "Human Fligth and Brain Drain") and add this in the Discussion. Available at: https://fragilestatesindex.org/country-data/.
4) The format should conform to the requirements of the journal, both in the presentation of references and figures.5) The text is interesting, but goes into little depth on brain drain theory. It rather samples different educational levels and concludes that it happens with more educated people. It is necessary to emphasize brain drain in the introduction and discussion.

Author Response

(The authors gave the same response as above.)

Round 2

Reviewer 2 Report

General remarks

The manuscript has improved. Not all my recommendations were met. I particularly regret the maintenance of the title, all the more so since, as the authors themselves acknowledge, this is a very specific case -- “[...] the sample only includes residents of Hanoi city, [...]” (page 15: 448-449), -- which, in my opinion, will be difficult to generalize to all developing countries. Therefore, I regret not agreeing with the authors in stating that “[...] the examined phenomenon and the examining methodology are applicable in developing countries in general.” (coverletter).

Author Response

Thank you for your comment! Please kindly check the attached file.

Reviewer 3 Report

Great effort has been made to modify what has been requested. At the final stage I recommend the editor to limit self-citation.

Author Response

(The authors gave the same response as above.)
